

# Effects of poly (ADP-ribose) polymerase-1 (PARP-1) inhibition on sulfur mustard-induced cutaneous injuries *in vitro* and *in vivo*

Feng Liu, Ning Jiang, Zhi-yong Xiao, Jun-ping Cheng, Yi-zhou Mei, Pan Zheng, Li Wang, Xiao-rui Zhang, Xin-bo Zhou, Wen-xia Zhou and Yong-xiang Zhang

Beijing Institute of Pharmacology and Toxicology, Beijing, China

## ABSTRACT

Early studies with first-generation poly (ADP-ribose) polymerase (PARP) inhibitors have already indicated some therapeutic potential for sulfur mustard (SM) injuries. The available novel and more potential PARP inhibitors, which are undergoing clinical trials as drugs for cancer treatment, bring it back to the centre of interest. However, the role of PARP-1 in SM-induced injury is not fully understood. In this study, we selected a high potent specific PARP inhibitor ABT-888 as an example to investigate the effect of PARP inhibitor in SM injury. The results showed that in both the mouse ear vesicant model (MEVM) and HaCaT cell model, PARP inhibitor ABT-888 can reduce cell damage induced by severe SM injury. ABT-888 significantly reduced SM induced edema and epidermal necrosis in MEVM. In the HaCaT cell model, ABT-888 can reduce SM-induced $NAD^+$/ATP depletion and apoptosis/necrosis. Then, we studied the mechanism of PARP-1 in SM injury by knockdown of PARP-1 in HaCaT cells. Knockdown of PARP-1 protected cell viability and downregulated the apoptosis checkpoints, including p-JNK, p-p53, Caspase 9, Caspase 8, c-PARP and Caspase 3 following SM-induced injury. Furthermore, the activation of AKT can inhibit autophagy via the regulation of mTOR. Our results showed that SM exposure could significantly inhibit the activation of Akt/mTOR pathway. Knockdown of PARP-1 reversed the SM-induced suppression of the Akt/mTOR pathway. In summary, the results of our study indicated that the protective effects of downregulation of PARP-1 in SM injury may be due to the regulation of apoptosis, necrosis, energy crisis and autophagy. However, it should be noticed that PARP inhibitor ABT-888 further enhanced the phosphorylation of H2AX (S139) after SM exposure, which indicated that we should be very careful in the application of PARP inhibitors in SM injury treatment because of the enhancement of DNA damage.

Corresponding authors
Wen-xia Zhou, pharmacare@126.com
Yong-xiang Zhang, zyxbigb@126.com

## INTRODUCTION

Sulfur mustard (SM), a chemical warfare vesicant agent, has been used in several military conflicts, including World Wars I and II and the Iran-Iraq War (*Balali-Mood & Hefazi, 2006*). After World War II, SM has also been one of the most predominant agents found

in the chemical weapons abandoned in China (*Hanaoka, Nomura & Wada, 2006*). Due to the potential use by terrorists against civilian populations, SM is currently considered to be a plausible global threat (*Smith et al., 1995*). The organs most affected by SM, skin, eyes and respiratory tract, are the organs that come in direct contact with the toxic liquid or vapor. Dermal exposure of SM results in severe chemical burns, affecting all layers of the skin, which produces subepidermal blisters, erythema and inflammation (*Balali-Mood & Hefazi, 2005*; *Ghanei et al., 2010*; *Naraghi, Mansouri & Mortazavi, 2005*). SM is a bifunctional alkylating compound that targets DNA, RNA, proteins, carbohydrates and lipids. Exposure to SM will cause DNA damage, apoptosis/necrosis and energy crisis in human cells. DNA damage is believed to be the most critical lesion after SM exposure (*Batal et al., 2015*; *Ludlum et al., 1994*; *Ludlum, Kent & Mehta, 1986*; *Yue et al., 2014*). The DNA damage induced by SM is considered to be the most important trigger of cell death in SM injury. In response to DNA damage, several key biochemical pathways important in protecting cells against injury and initiating repair processes are activated. PARP-1, a nicotine adenine dinucleotide ($NAD^+$)-dependent nuclear enzyme implicated in multiple DNA repair pathways (including single strand breaks, double strand breaks and base excision repair), has been shown to be activated upon SM exposure (*Hinshaw et al., 1999*; *Mangerich et al., 2015*; *Rosenthal et al., 1998*). PARP-1 uses $NAD^+$ as a substrate to catalyze covalent binding of poly (ADP-ribose) (PAR) to PARP-1 itself and some other nuclear proteins involved in DNA repair, inflammation, apoptosis/ necrosis and autophagy (*Sodhi, Singh & Jaggi, 2010*). Although PARP-1 activation had been considered to have a positive effect on cell survival by promoting DNA repair, overactivation of PARP-1 initiated by severe SM-induced DNA damage and the consequent depletion of $NAD^+$ exacerbated the cell death instead. The $NAD^+$ depletion could be prevented by pre-treatment with the PARP inhibitor (*Papirmeister et al., 1985*). However, little is known about the relationship between the apoptosis and autophagy pathways and the protective effect of PARP-1 inhibitors in SM-induced injury.

Research on animal models revealed that PARP inhibitors could reduce the pathological damage in the treatment of skin injuries (*Casillas et al., 2000*; *Cowan et al., 2003*; *Yourick, Clark & Mitcheltree, 1991*), indicating that PARP inhibitors might have protective effects on SM-induced injury, but the comprehensive pharmacodynamic evaluation of PARP inhibitors for SM injury treatment is still lacking. Actually, there is no further evidence supporting the protective effect of PARP inhibitors on SM injured animal models in nearly 10 years. *In vitro*, while the cellular $NAD^+$ depletion induced by SM injury could indeed be prevented by treatment with PARP inhibitors, the effect on total cell survival could not be observed (*Kehe et al., 2008*). It is worth noting that, theoretically, PARP inhibition can block DNA repair and increase cell death induced by alkylating agents (*Carey & Sharpless, 2011*; *Xiong et al., 2015*). Previous studies also showed that PARP inhibitors indeed delayed the DNA repair in SM injury (*Bhat, Benton & Ray, 2006*; *Bhat et al., 2000*). Taken together, the above studies indicated that the role of PARP inhibitors in SM injury is not fully understood, which is definitively worthy of more detailed investigation (*Debiak, Kehe & Burkle, 2009*).

In recent years, there is a new generation of PARP inhibitors available in cancer chemotherapy (*Coleman et al., 2015*; *Gunderson & Moore, 2015*; *Jones et al., 2015*; *Thomas et al., 2007*), which are much more potent than the classical ones. Whether these more specific and potent PARP inhibitors may also be beneficial in treating SM injuries remains to be shown. Therefore, in this study we use ABT-888 hydrochloride, a new generation of PARP inhibitor which is over 100 times more potent than 3-AB (the classical PARP inhibitor that was widely used in treatment of SM injury), as an example to evaluate its effects on immortalized keratinocytes (HaCaT cells) and mouse ear vesicant model (MEVM) treated with SM. Furthermore, we study the mechanism of PARP-1 in SM injury by knockdown of PARP-1 in HaCaT cells.

## MATERIALS AND METHODS

### Chemicals

SM (bis-[2-chloroethyl] sulfide, purity 97.6%) was provided by the Institute of Chemical Defense of CPLA (Chinese People's Liberation Army). ABT-888 hydrochloride (*Tian-Xiang et al., 2013*) was synthesized in our institute (purity 99%). The skeletal formula for compound ABT-888 hydrochloride is shown. Other reagents were obtained from Sigma unless otherwise mentioned.

ABT-888 hydrochloride
2-[(S)-2-methylpyrrolidin-2-yl]-
1H-benzimidazole-4-carboxamide

### Cells

Immortalized epithelial keratinocytes (HaCaT) was a spontaneously transformed aneuploid immortal keratinocyte cell line from adult human skin (*Boukamp et al., 1988*), which was purchased from the China Center for Type Culture Collection (CCTCC) and cultured in Dulbecco's Modified Eagle Medium/Ham's F12 (DMEM/F12).

### Detection of PARP-1 enzyme inhibitory activity

The inhibitory activity of the PARP-1 enzyme was detected using the Trevigen's Homogeneous PARP Inhibition Assay Kit. This assay was performed according to the manufacturer's instructions.

### Animals

Twenty-nine adult male Kunming (KM) mice (18–22 g) were obtained from the Beijing Vital River Laboratory. The animals were acclimatized for one week before being randomly assigned to experimental groups for use in the experimental studies. The mice were housed in an animal house ($26 \pm 2$ °C) and were provided with water and food ad libitum

throughout the experiment. Briefly, a total of 29 mice were randomly divided into 5 groups: (i) untreated control ($n = 5$), (ii) 0.16 mg SM/ear ($n = 5$), (iii) 0.64 mg SM/ear ($n = 7$), (iv) 0.16 mg SM/ear + ABT-888 ($n = 5$), and (v) 0.64 mg SM/ear + ABT-888 ($n = 7$). The experiments were carried out following protocols approved by the Anima Ethics Committee, Beijing Institute of Pharmacology and Toxicology. The study was conducted according to the Care and Use Guide for Laboratory Animals by the NIH and was approved by the Bioethics Committee of the Beijing Institute of Pharmacology and Toxicology (No. 80-23).

## Exposure of HaCaT cells to SM

The exponentially growing HaCaT cells were seeded in either plates or dishes. Before the exposure to SM, the culture medium was discarded and then 100 or 1,000 μM SM were added to the plates. After 30 min, the agent was removed and the cells were washed with phosphate buffered saline (PBS). DMEM/F12 (with 10% fetal calf serum) alone or with ABT-888 was added until cells were analyzed as described.

## Cell viability assay

Cell viability was quantified using the Cell Counting Kit-8 (CCK-8) (Dojindo). This assay is based on Dojindo's highly water-soluble tetrazolium salt. WST-8 [2-(2-methoxy-4-nitrophenyl)-3-(4-nitrophenyl)-5-(2,4-disulfophenyl)-2H-tetrazolium, monosodium salt] is reduced by dehydrogenases in cells to give an orange, water-soluble formazan dye. The amount of the formazan dye generated by dehydrogenases in cells is directly proportional to the number of living cells. Briefly, exponentially growing HaCaT cells were seeded in 96-well plates at a density of 10,000 cells/well. 6 h or 24 h after exposure to SM and the administration of ABT-888, the CCK-8 reagent was added as recommended by the supplier.

## pADPr immunofluorescence

HaCaT cells were seeded in MatTek glass bottom culture dishes and treated with SM and ABT-888. 6 h after exposure to SM and the administration of ABT-888, the cells were washed in PBS and fixed in ice cold 100% methanol for 10 min. The images were obtained by confocal microscopy. The primary antibody used was the anti-pADPr antibody (Abcam) and the secondary antibody was AlexaFluor 488 goat anti-mouse IgG (Molecular Probes). The antibody was dissolved in PBS containing 5% bovine serum albumin (BSA). Images were obtained using a Zeiss LSM 510 META confocal microscope. The mean fluorescence intensity for pADPr was calculated for each individual nucleus using the PI-marked DNA as a nuclear marker. Approximately 30 cells from three different images were analyzed with the ImageJ program.

## Acumen

HaCaT cells were seeded in 96-well plates and treated with SM. After 6 h of exposure to SM, the cells were washed with PBS and fixed with 4% paraformaldehyde for 15 min and permeabilized in 100% pre-cooled methanol for 5 min. The cells were then blocked in 5% BSA and incubated with the anti-pADPr antibody (Abcam) for 1 h followed by labeling with AlexaFluor 488 goat anti-mouse IgG (Molecular Probes) for 1 h. Then, the cells

were stained with 0.3 $\mu$ g/well Hoechst 33342 (Sigma) in PBS for 30 min. The plates were scanned on an Acumen eX3 laser scanning cytometer (TTP LabTech, Melbourne, UK), and the pADPr/nuclear Total Fluorescence Intensity was calculated using the Acumen eX3 software.

## Western blot

Briefly, the cells were washed with cold PBS and lysed on ice for 30 min in a lysis buffer containing 1× protease inhibitor cocktail (Roche). The cell lysates were centrifuged at 12,000× g for 30 min, and the supernatants were collected. The proteins were separated by SDS-PAGE and transferred to NC membranes (Bio-Rad), which were blocked for 2 h in PBS with 5% BSA and 0.1% Triton X-100. Subsequently, the membranes were incubated with primary antibodies overnight at 4 °C followed by incubation with the appropriate horseradish peroxidase (HRP)-labeled secondary antibodies for 2 h at room temperature. The immunoreactive bands were detected using SuperSignal West Pico Chemiluminescent detection reagents (Thermo SCIENTIFIC). The rabbit anti-PARP-1 antibodies (Cell Signaling Technology), rabbit anti-Actin antibody (Abcam), mouse anti-pADPr antibody (Abcam) and mouse anti-GAPDH antibody (Cell Signaling Technology) were used as primary antibodies.

## Detection of $\gamma$-H2AX

HaCaT cells were seeded in 6-well plates. Six hours or 24 h after SM exposure and ABT-888 administration, the cells were harvested, permeabilized and fixed. The primary antibody was the anti-gamma H2A.X (phospho S139) antibody (Abcam) and the secondary antibody was anti-rabbit IgG-FITC (Santa Cruz Biotechnology). A FACSCalibur flow cytometer (Becton, Dickinson and Company, BD) was used to detect the fluorescence intensity.

## NAD$^+$ quantification

Intracellular NAD$^+$ levels were measured using the NADH/NAD Quantification kit (Sigma) according to the manufacturer's instructions. Briefly, HaCaT cells were seeded and harvested 6 h or 24 h after treatment. Next, cells were extracted with NADH/NAD Extraction Buffer. Total NAD$^+$ was detected in a 96-well plate following the protocol instructions. The color reaction was read with a 450 nm filter (Enspire 2300, PERKINELMER, PE).

## Determination of intracellular ATP

The concentration of intracellular ATP was determined using a bioluminescence ATP assay kit (Beyotime) (Sigma) according to the manufacturer's instructions. Briefly, HaCaT cells were seeded and harvested 6 h or 24 h after treatment. The cells were extracted with lysis buffer and an ATP detection working solution was added to each well of a black 96-well plate, followed by incubation for 3 min at room temperature. Then, the cell lysate samples were added to the wells and the luminescence was measured immediately (Enspire 2300, PE).

## Measurement of caspase 3/7 activity

Caspase 3/7 activities were measured using the Caspase-Glo 3/7 assay Kit (Promega, Corp.) following manufacturer's instruction. 6 h or 24 h after treatment, the HaCaT cells

were lysed, and the substrate cleavage by caspase 3/7 was measured from the generated luminescent signal (Enspire 2300, PE).

## Quantification of apoptotic and necrotic cell death

Apoptosis and necrosis were measured by annexin/propidium iodide (PI) double staining. Six hours or 24 h after treatment, HaCaT cells were harvested and washed with PBS. Then, the cells were re-suspended in 500 µl binding buffer (10 mM HEPES, 140 mM NaCl, 2.5 mM CaCl2, pH 7.4) and stained with 5 µl annexin V-FITC and 5 µl PI (Invitrogen). The cells were incubated in a dark at room temperature for 15 min according to the manufacturer's instructions and subjected to flow cytometry.

## Transient transfection of PARP-1shRNA and selection of stable transfectants

The RNAi lentivirus that was used to specifically interfere with the PARP-1 gene was designated PARP-1 shRNA (5′-CCGGCGACCTGATCTGGAACATCAACTCGAGTTGATG TTCCAGATCAGGTCGTTTTT-3′), and the control shRNA, which was used as the negative control, was designated NC. To establish stable cell lines with knockdown of PARP-1, Lv-shPARP-1 was transfected into HaCaT cells. At 24 h after transfection, the cells were portioned into new plates and subjected to selection with 3 µg/ml Puromycin (Invitrogen) for 10 days. Independent colonies were isolated and confirmed by western blotting and PCR. Control colonies stably transfected with Lv-shCon were also generated in parallel. Lv-shPARP-1 and Lv-shCon were constructed by BioWit Technologies Co., Ltd (Shenzhen, China).

## RNA isolation and RT-PCR

Total RNA from cultured cells was extracted with Trizol reagent (Invitrogen) according to the manufacturer's instructions. RNA (1 µg) was converted to cDNA using the PrimeScript RT reagent kit (Takara Biotechnology). Two sets of primers were used for PCR: GAPDH forward, 5′-AGGTGAAGGTCGGAGTCAAC-3′ and reverse, 5′-CGCTCCTGGAA GATGGTGAT-3′; and PARP-1 forward, 5′-GAGCATCCCCAAGGACTCG-3′ and reverse, 5′-CCGCTGTCTTCTTGACTTTC-3′. The mRNA expression levels were determined using the SYBR Premix Ex Taq II kit (Takara Biotechnology). The relative mRNA expression of PARP-1 was calculated using the 2-$\Delta\Delta$Ct method.

## Luminex assay

At 6 h or 24 h after SM exposure, HaCaT cells were lysed. The cell suspension was transferred into a centrifuge tube and gently rocked for 10–15 min at 4 °C. Then, the particulate matter was removed by filtration using EMD Millipore filters. The Phospho-JNK (Thr183/Tyr185), Phospho-p53 (ser46), Active Caspase 9 (Asp315), Active Caspase 8 (Asp384), c-PARP (p89), Phospho-AKT (Thr308) and Phospho-mTOR (Ser2448) protein levels were determined with the Luminex assays using a MILLIPLEX MAP (Millipore Corp) according to the manufacturer's protocols. The data were collected using a Luminex 200 analyzer (Luminex Corp).

## Mouse ear vesicant model

In the mouse ear vesicant model (MEVM), five microliters (32 mg/ml and 128 mg/ml) of SM in propylene glycol was applied to the medial surface of the right ears of KM mice. Propylene glycol was applied to the medial surface of the right ear of KM mice in the control group. ABT-888 was administered (i.p.) at dose of 200 mg/kg once 30 min before or 10 min after the SM exposure. The injury was measured from the edema response. The ear edema for each animal was initially expressed as a percentage of the increase in relative ear weight (REW) to normal and was calculated from the difference in ear weights between the right and left ears using the following formula:

$$\% \, REW = \frac{SM \, (or \, ABT-888+SM) \, exposed \, ear \, weight \, (right \, ear) - Vehicle \, control \, ear \, weight \, (left \, ear)}{Vehicle \, control \, ear \, weight \, (left \, ear)} \times 100.$$

The effect of the drug (modulation of edema) was expressed as the percentage in reduction from the positive control (SM-exposed only) and was calculated from the difference in the mean % REW between the ABT-888 + SM group and the SM only group (positive control) using the following formula (*Babin et al., 2000*):

$$\% \, Reduction \, from \, model = \frac{(Mean\% \, REW \, ABT-888+SM) - (Mean\% \, REW \, SM)}{Mean\% \, REW \, SM} \times 100.$$

## Histopathology

Immediately after obtaining the tissue weight, skin punches were fixed in 10% neutral buffered formalin (NBF). All NBF specimens were embedded in paraffin, sectioned and stained with hematoxylin and eosin (H& E) for evaluation.

The histopathological endpoints described in epidermal necrosis (EN; inner surface of the SM treated ear) were given the following severity scores as previously described (*Babin et al., 2000*): (0) no lesion or change; (1) change in <5% of the entire tissue section; (2) change in 10–40% of the entire tissue section; (3) change in 50–80% of the entire tissue section; (4) change in >90% of the entire tissue section. For example, a severity score of 2 (change in <10%–40% of the entire tissue section) indicates 10%–40% of the epidermal cells are necrotic cells in a 400× field. Histopathology was evaluated in a blind experiment and the scores were reported as the mean and standard deviation of each group.

## Statistical analysis

All values were expressed as the means ± SEM. For the evaluation of statistical significance, a two-tailed Student's $t$-test was performed. Bonferroni test was used for post hoc comparison. $P < 0.05$ was considered statistically significant.

## RESULTS

### PARP-1 activation increased in HaCaT cell following SM treatment, which could be prevented by ABT-888

SM is a highly reactive bifunctional alkylating agent that induces DNA damage. DNA damage activates PARP-1, which catalyzes the transfer of ADP-ribose units from

nicotinamide adenine dinucleotide (NAD$^+$) to PARP-1 itself and a range of other nuclear proteins. To determine the activation of PARP-1 and the PARP inhibitory activity of ABT-888 in SM treated HaCaT cells, the expression of pADPr was detected by immunofluorescence and western blotting. The increase of pADPr, which represented the activation of PARP-1, was observed after 6 h of exposure to SM at concentrations of either 100 µM or 1,000 µM. The activation of PARP-1 under 1,000 µM SM exposure was much greater than under 100 µM SM exposure. Immediate application of ABT-888 after SM exposure could significantly decrease the expression of pADPr, especially under 1,000 µM SM exposure. In addition, ABT-888 alone group and control group did not show any significant difference in the expression of pADPr (Figs. 1A–1E). These results suggested that the SM injury could cause PARP-1 activation, which could be marked inhibited by the potent PARP inhibitor ABT-888. The PARP-1 inhibitory effect of ABT-888 was also confirmed using the Trevigen's Homogeneous PARP Inhibition Assay Kit (Fig. 1F).

## PARP inhibitor ABT-888 had a protective effect in HaCaT cells after severe SM injury

In HaCaT cells, at 6 h post-treatment by ABT-888, cell viability was significantly increased under 1,000 µM SM exposure (Fig. 2A), whereas ABT-888 did not protect cell viability under 100 µM SM exposure. Moreover, the addition of ABT-888 no longer showed the protective effect at 24 h post SM exposure (Fig. 2B). ABT-888 alone group and control group did not show any significant difference in cell viability.

## PARP inhibitor ABT-888 had a protective effect in mouse ear vesicant model after severe SM injury

In the SM-induced mouse ear vesicant model, we also observed protective effect of ABT-888. ABT-888 showed a protective effect in severe SM injury. The structural alterations of the skin in the mouse ears post SM exposure were observed. Exposure to 0.16 mg SM/ear and 0.64 mg SM/ear resulted in moderate to severe edema and epidermal necrosis at 24 h after SM exposure. Twenty-four hours post-exposure to 0.16 mg SM/ear and 0.64 mg SM/ear, the dermis inflammatory cell infiltration and reticular degenerative changes associated with basal cell necrosis of the epidermis were observed, along with hypereosinophilic cytoplasms and nuclear pyknosis. Epidermal cells exposed to 0.16 mg SM/ear showed pyknotic nuclei, but ABT-888 did not show any protective effect. More pyknotic nuclei of epidermal cells were observed in the mouse ear skin exposed to 0.64 mg SM/ear than that exposed to 0.16 mg SM/ear and ABT-888 could significantly reduce the number of pyknotic nuclei (Fig. 3A).

The increase in relative ear weight (REW) to normal represented the ear edema. The REW and epidermal necrosis (EN) scores for the right ear of each animal in the control group were zero (Table S3). The REW and EN scores of all SM groups were significantly different from those of the control group, and the scores of the group exposed to 0.64 mg SM/ear (REW 204%, EN 3.9) were higher than those of the group exposed to 0.16 mg SM/ear (REW 154%, EN 2.2). ABT-888 did not reduce the REW and EN scores in the group exposed to 0.16 mg SM/ear, but ABT-888 reduced the REW (approximately 26%)and EN

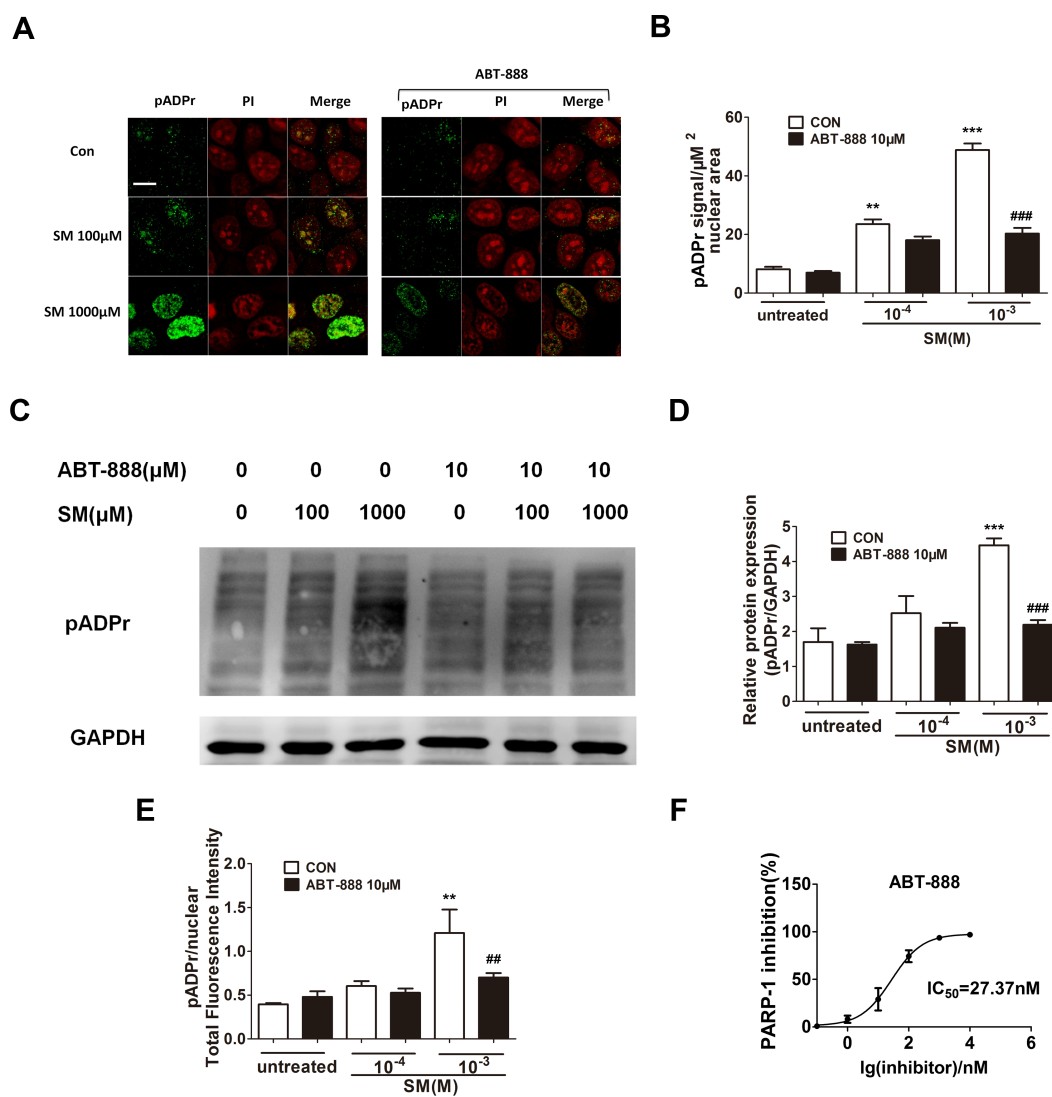

**Figure 1  Effect of PARP-1 inhibitors on pADPr content (which represents PARP-1 activity) in SM-treated HaCaT cells.** Cultured HaCaT cells were treated with various concentrations of SM for 6 h. The level of pADPr protein was determined by immunofluorescence (A and B), western blotting (C and D) and Acumen (E), respectively. (A and B) The scale shown in the upper left panel (bar = 20 μm) was the same for all panels. (C and D) The results of the western blots were normalized to the levels of GAPDH and then presented as the fold of the control levels. (E) The results of the Acumen analysis were normalized to the nuclear total fluorescence intensity. **$P < 0.01$, ***$P < 0.001$ vs. untreated group. ##$P < 0.01$, ###$P < 0.001$ vs. SM-treated group. (F) The PARP-1 inhibitory effect of ABT-888 was also confirmed at enzyme level. (Each data point represents three data points).

scores (approximately 40%) in the group exposed to 0.64 mg SM/ear (Figs. 3B and 3C). The results were similar when ABT-888 was applied either after or before SM treatment (Table S10). In addition, the ABT-888 alone group and control group did not show any significant difference (Table S10).

**A**

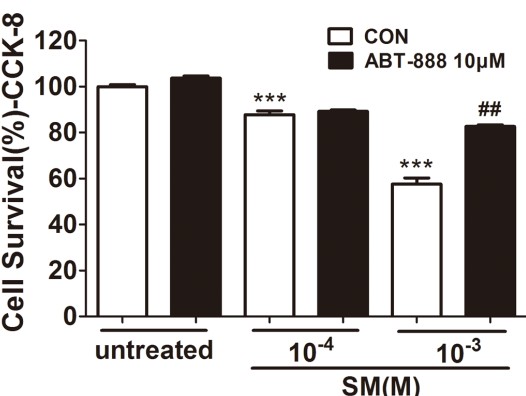

**B**

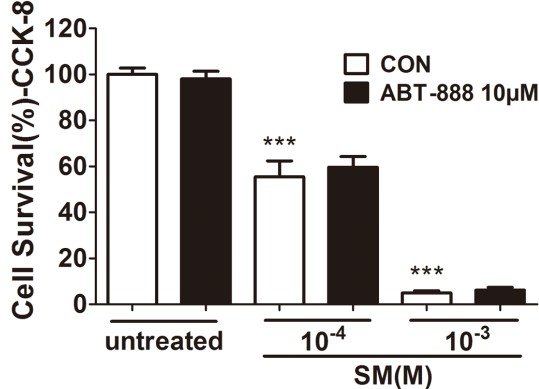

**Figure 2** **Effect of PARP inhibitors on cell survival in SM-treated HaCaT cells.** Cell viability was quantified using the Cell Counting Kit-8 6 h (D) and 24 h (E) after SM exposure and the treatment of ABT-888. ***$P < 0.001$ vs. untreated group. ##$P < 0.01$ vs. SM-treated group. All the data were presented as means $\pm$SEM ($n = 6$).

## The PARP inhibitor ABT-888 could prevent the reduction of NAD$^+$ and ATP in HaCaT cell after SM exposure

SM has been shown to activate the enzymatic activity of PARP-1 by decreasing the NAD$^+$ concentration in either human skin grafts (*Gross et al., 1985*) or keratinocytes *ex vivo* (*Rosenthal et al., 1998*). Severe SM-induced DNA damage over-activates PARP-1, leading to the depletion of NAD$^+$ and ATP. To determine the NAD$^+$ and ATP concentration after SM exposure and the effect of ABT-888, HaCaT cells were exposed to SM and the concentrations of intracellular NAD$^+$ andATP were measured 6 and 24 h later. SM exposure could decrease NAD$^+$ and ATP levels (Figs. 4A–4D). The ABT-888 alone group and control group did not show any significant difference in NAD$^+$ and ATP concentration. Immediate application of ABT-888 after SM exposure could prevent the reduction of NAD$^+$ and ATP.

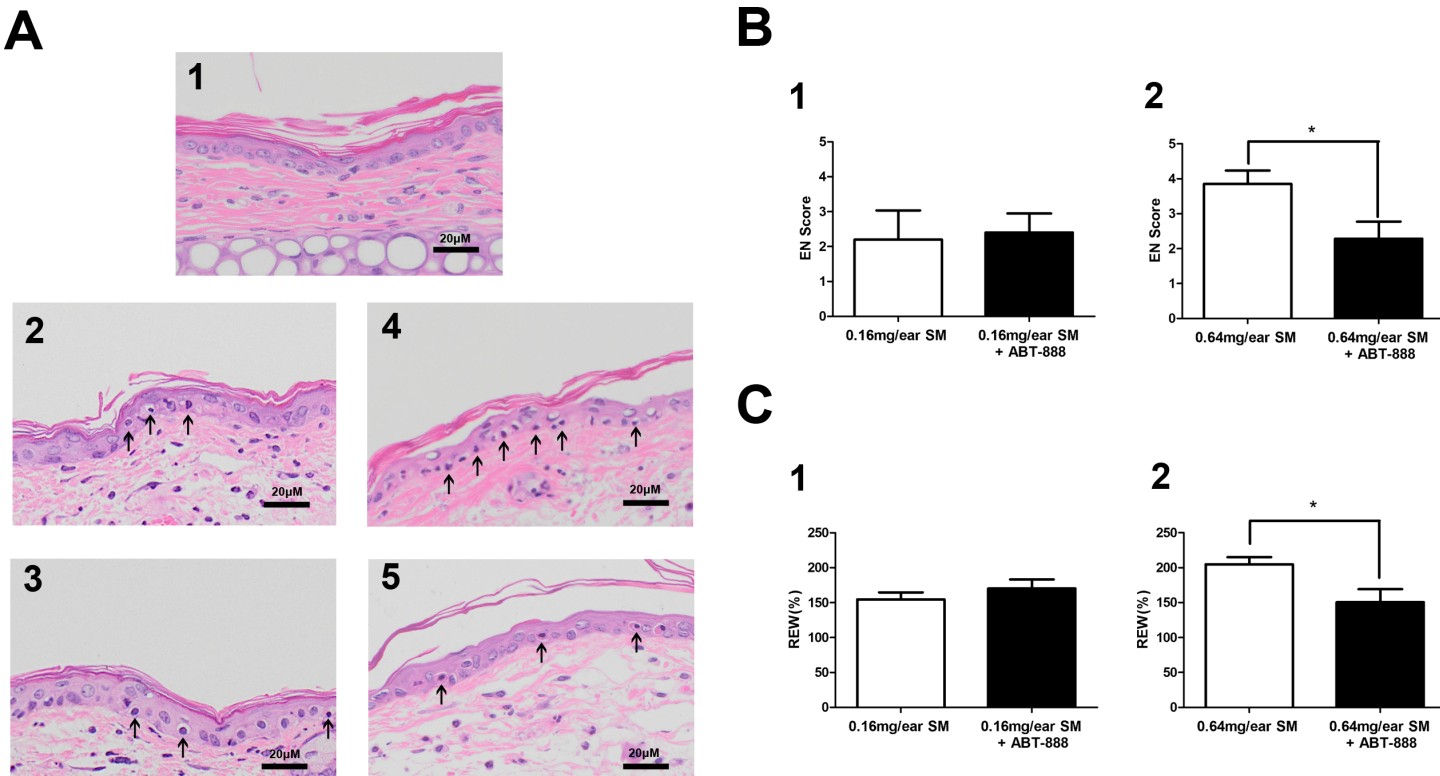

**Figure 3** **Effect of PARP-1 inhibitors on relative ear weight and histopathological change in the SM-treated mouse ear vesicant model.** (A) Hematoxylin and eosin stains of the mouse ear skin showed that the PARP inhibitor ABT-888 did not have a protective effect against pathological damage in mice exposed to 0.16 mg SM/ear, but ABT-888 produced reductions in pathological damage in mice exposed to 0.64 mg SM/ear. (A1) The medial surface of a normal ear from the control group. (A2) The medial surface of an ear from the 0.16 mg SM/ear exposure group, showing epidermal necrosis (pyknotic nuclei, arrowhead). (A3) The medial surface of an ear from the 0.16mg SM/ear exposure + ABT-888 group. ABT-888 showed no protective effect. (A4) The medial surface of an ear from the 0.64 mg SM/ear exposure group. Epidermal necrosis (arrowhead) was progressively more severe. (A5) The medial surface of an ear from the 0.64 mg SM/ear exposure + ABT-888 group. ABT-888 could reduce reticular degenerative changes in the dermis, hypereosinophilic cytoplasms of the epidermis necrosis (arrowhead). The scale shown in the lower right panel (bar = 20 μm) is the same for all panels. (B and C) ABT-888 significantly reduced edema (REW, approximately 26%) of the ear and epidermal necrosis (EN score, approximately 40%) in MEVM in the group exposed to 0.64 mg SM/ear, but showed no protective effect in the group exposed to 0.16 mg SM/ear. ABT-888 was administered (i.p.) 30 min before the SM exposure. *$P < 0.05$ vs. 0.64 mg SM/ear. All the data are presented as means ±SEM ($n = 5$–7).

### The PARP inhibitor ABT-888 suppressed SM-induced apoptosis and necrosis in HaCaT cells

It has been known that SM-induced DNA injury causes PARP activation, which may lead to necrosis or apoptosis. PARP-1 has been shown to participate in apoptotic and necrotic cell death pathways. Therefore, we used flow cytometry to analysis HaCaT cells doubly stained for annexin V and PI to observe the SM-induced apoptosis and necrosis and to assess the effect of ABT-888. The results showed that only 1,000 μM SM caused HaCaT cell apoptosis and necrosis after SM exposure. ABT-888 significantly decreased the percentage of apoptotic and necrotic cells (Figs. 5E and 5F).

Thus, we next assessed whether SM activates the caspase-PARP pathway, which is the main executor of apoptotic processes. Caspase-3/7 is responsible for the cleavage of PARP-1 during apoptosis. Therefore, the caspase-3/7 activity was measured after treatment

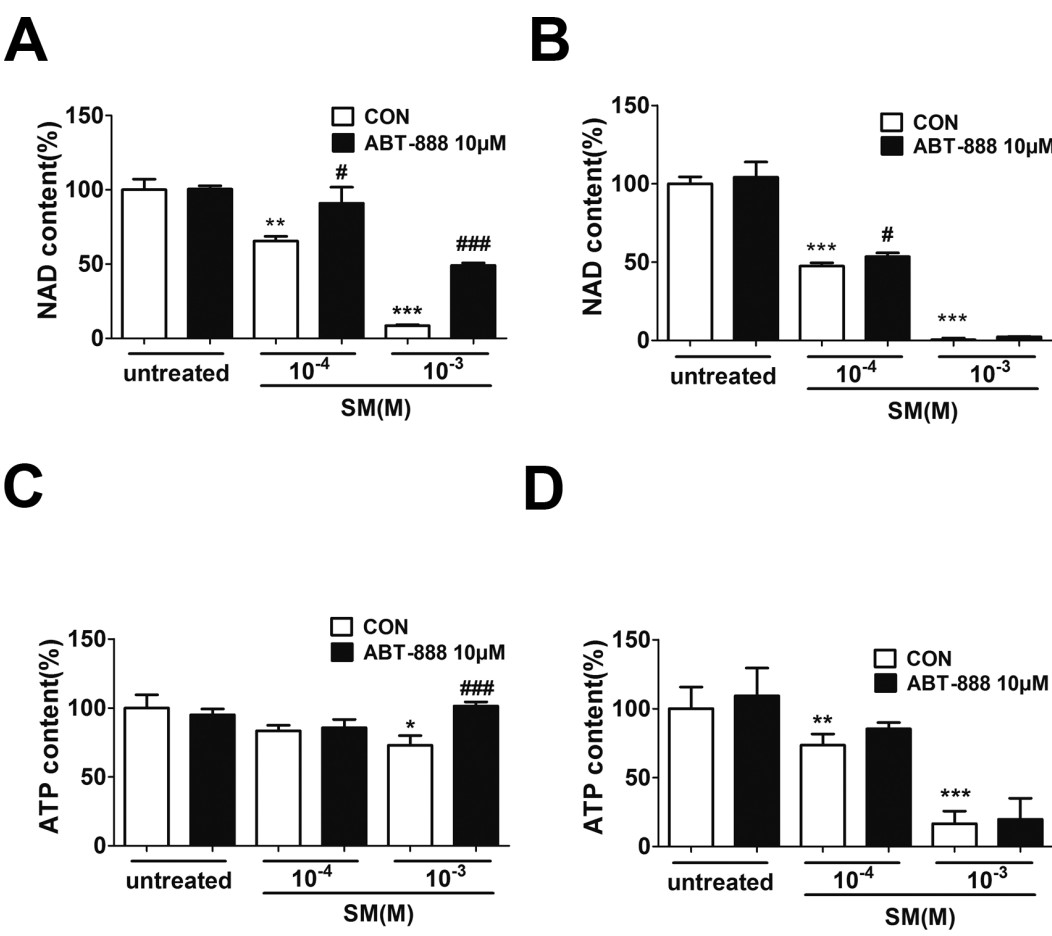

**Figure 4** **The effects of the PARP inhibitor on NAD$^+$/ATP content in SM-treated HaCaT cells.** Intracellular NAD$^+$ levels were measured 6 h (A) and 24 h (B) after SM exposure and the treatment of ABT-888. Intracellular ATP levels were measured 6 h (C) and 24 h (D) after SM exposure and the treatment of ABT-888. The values are presented as means ±SEM, $n = 3$ or 6. $*P < 0.05$, $**P < 0.01$, $***P < 0.001$ vs. untreated group. $^\#P < 0.05$, $^{\#\#\#}P < 0.001$ vs. SM-treated group.

of HaCaT cells with 100 or 1,000 μM SM for 6 h or 24 h. At 6 h following 100 or 1,000 μM SM exposure, an increase in caspase-3/7 activity was observed. ABT-888 decreased caspase 3/7 activity in cells exposed to 1,000 μM SM but increased caspase 3/7 activity in cells exposed to 100 μM SM (Fig. 5A). At 24 h following SM exposure, ABT-888 decreased the caspase 3/7 activity in HaCaT cells (Fig. 5B).

When apoptosis occurs, PARP-1 is cleaved by activated caspase 3/7 into p89 and p24 fragments. To observe the effects of ABT-888 on apoptosis at different doses and incubation times with SM, a western blot was performed to detect c-PARP (cleavage of PARP-1, an apoptosis-specific p89 fragment). The results showed that SM could significantly increase the expression of c-PARP. At 6 h after 1,000 μM SM exposure, ABT-888 decreased the expression of c-PARP. However, ABT-888 showed no significant effect on c-PARP levels in cells exposed to 100 μM SM (Fig. 5C). However, ABT-888 had no significant protective effects on c-PARP levels at 24 h after SM exposure (Fig. 5D). In addition, ABT-888 alone

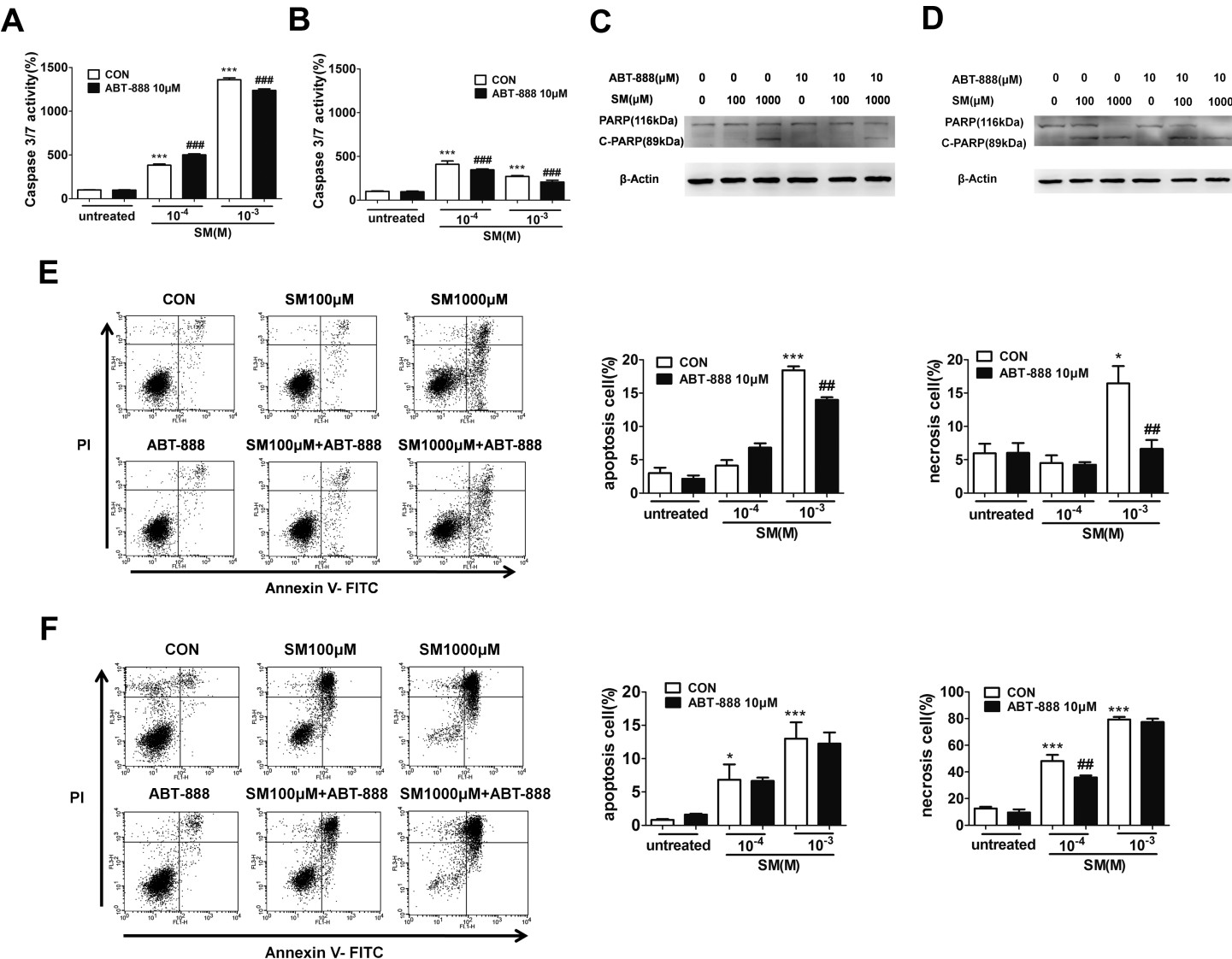

**Figure 5** **The effects of the PARP inhibitor ABT-888 on apoptosis/necrosis in SM-treated HaCaT cells.** Caspase 3/7 activity was measured 6 h (A) and 24 h (B) after SM exposure and the treatment of ABT-888. Total cell lysates were prepared and analyzed for protein expression by western blotting using anti-PARP-1 and cleavage of PARP-1 antibody 6 h (C) and 24 h (D) after SM exposure and the treatment of ABT-888. $\beta$-Actin was used as a control to ensure equal protein loading. Apoptosis and necrosis were analyzed by flow cytometry 6 h (E) and 24 h (F) after SM exposure and the treatment of ABT-888. The frames were divided into four quadrants: Annexin V−/PI− normal cells are in quadrant I; Annexin V+/PI− apoptotic cells are in quadrant II; PI+ necrotic cells are in quadrant III/IV. The values are presented as means ± SEM, $n = 3$ or 6. *$P < 0.05$, **$P < 0.01$, ***$P < 0.001$ vs. untreated group. ##$P < 0.01$, ###$P < 0.001$ vs. SM-treated group.

group and control group did not show any significant difference in caspase-3/7 activity, c-PARP levels, percentage of apoptotic and necrotic cells.

## PARP-1 knockdown protected HaCaT cells from SM-induced cell death

To further investigate the mechanism of PARP-1 on the toxicity of SM, we used RNA interference to knock down the endogenous PARP-1 in HaCaT cells. To examine the effect of PARP-1 knockdown on the mRNA level, the total mRNA from control and

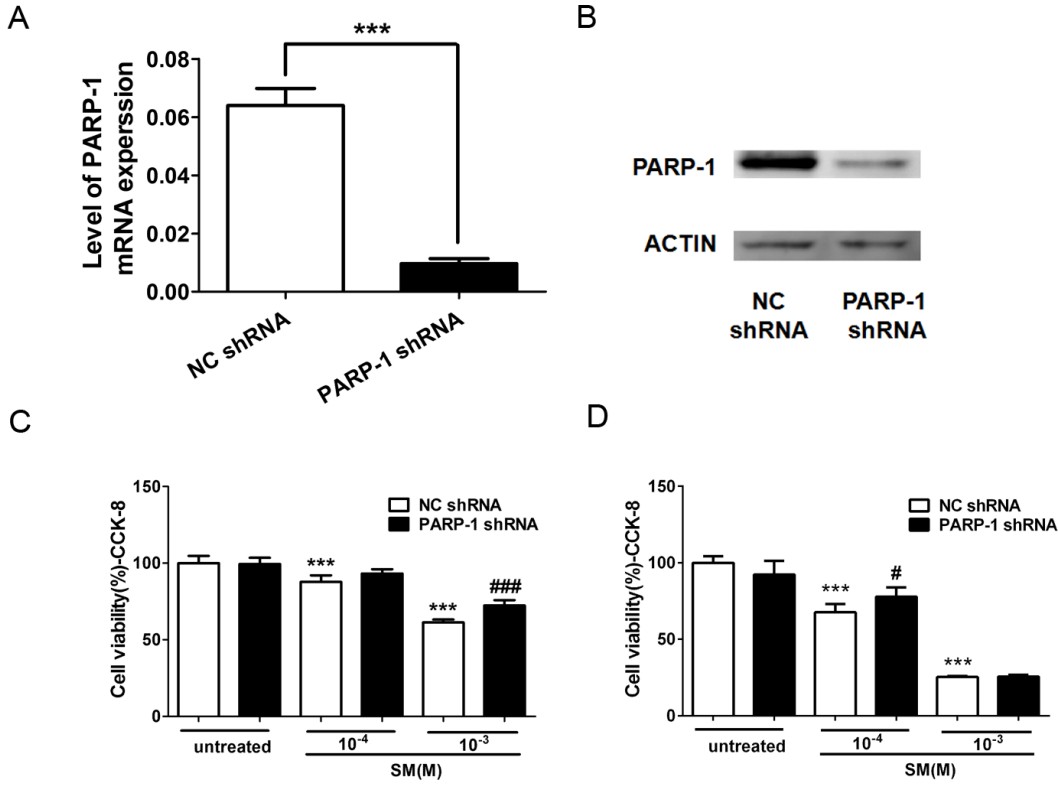

**Figure 6 Knockdown of PARP-1 suppressed the toxicity of SM at the level of cell viability.** To evaluate whether PARP-1 was involved in the toxicity of SM, Lv-shPARP-1 and Lv-shCon were transfected into HaCaT cells; then, stable transfectants were selected. The efficacy for PARP-1 knockdown was determined by RT-PCR (A) and western blotting (B). The control and PARP-1 knockdown HaCaT cells were treated with 0, 100, or 1,000 μM SM. Subsequently, the cell viability was measured 6 h (C) and 24 h (D) after exposure to SM. The results are presented as means ± SEM determined from three independent experiments. ***$P < 0.001$ vs. untreated group. #$P < 0.05$, ###$P < 0.001$ vs. SM-treated group.

PARP-knockdown HaCaT cells was isolated and processed for RT-PCR. The relative mRNA expression of PARP-1 was calculated. The results showed that PARP-1 shRNA significantly decreased the levels of PARP-1 mRNA by 90% compared to the level in the control (Fig. 6A). The level of PARP-1 protein was also significantly decreased by RNA interference (Fig. 6B).

To study the effect of PARP-1 knockdown in the toxicity of SM, we examined the cell viability in PARP-1-knock-down and control HaCaT cells exposed to SM using the CCK-8 kit. The results showed that 100 or 1,000 μM SM exposure decreased the cell viability 6 h and 24 h after SM treatment. The downregulation of PARP-1 significantly increased cell viability compared with that of the control SM-treated cells (Figs. 6C and 6D), suggesting that PARP-1 knockdown suppressed the toxicity of SM at the level of cell viability.

## PARP-1 knockdown suppressed the SM-induced apoptosis checkpoint signals in HaCaT cells

In HaCaT cells treated with NC shRNA, the protein levels of phospho-JNK (Thr183/Tyr185), phospho-p53 (ser46), active Caspase 9 (Asp315), active Caspase 8

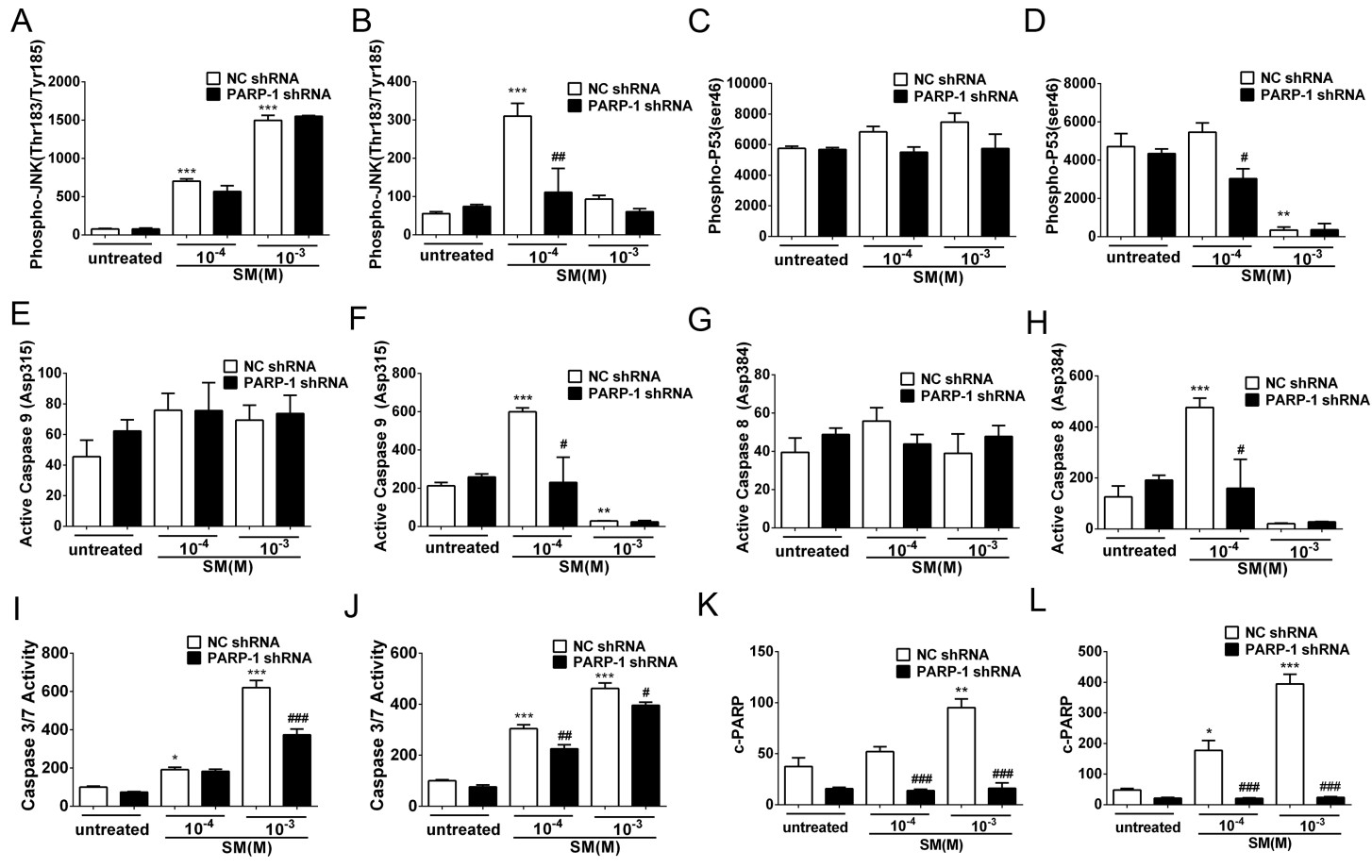

**Figure 7  SM-induced activation of apoptosis checkpoint signals was suppressed by PARP-1 knockdown in HaCaT cells.** PARP-1-knockdown and control HaCaT cells were treated with 100 μM or 1,000 μM SM. At 6 h and 24 h after exposure to SM, the cells were harvested for the detection of the apoptosis checkpoint signals. The protein levels of phospho-JNK (Thr183/Tyr185) 6 h (A) and 24 h (B) after exposure to SM, phospho-p53 (ser46) 6 h (C) and 24 h (D) after exposure to SM, active Caspase 9 (Asp315) 6 h (E) and 24 h (F) after exposure to SM, active Caspase 8 (Asp384) 6 h (G) and 24 h (H) after exposure to SM, and c-PARP (p89) 6 h (K) and 24 h (L) after exposure to SM were determined using Luminex assays. The caspase 3/7 activity 6 h (I) and 24 h (J) after exposure to SM was measured using the Caspase-Glo 3/7 assay kit. The results are presented as means ±SEM, as determined from three independent experiments. *$P < 0.05$, **$P < 0.01$, ***$P < 0.001$ vs. untreated group. #$P < 0.05$, ##$P < 0.01$, ###$P < 0.001$ vs. SM-treated group.

(Asp384), c-PARP (p89) and Caspase 3/7 activity were enhanced in response to SM stimulation (Figs. 7A–7L). However, 24 h after exposure to 1,000 μM SM, the protein levels of phospho-JNK (Thr183/Tyr185) (Fig. 7B), phospho-p53 (ser46) (Fig. 7D), active Caspase 9 (Asp315) (Fig. 7F), and active Caspase 8 (Asp384) (Fig. 7H) were extremely low, possibly because these early apoptosis signals could not be detected during the late apoptosis. In the HaCaT cells treated with PARP-1 shRNA, all of the apoptosis checkpoint signals mentioned above was downregulated compared to the cells treated with NC shRNA (Figs. 7A–7L). These results suggested that the silencing of PARP-1 suppressed the SM-induced apoptosis checkpoint signals in the HaCaT cells.

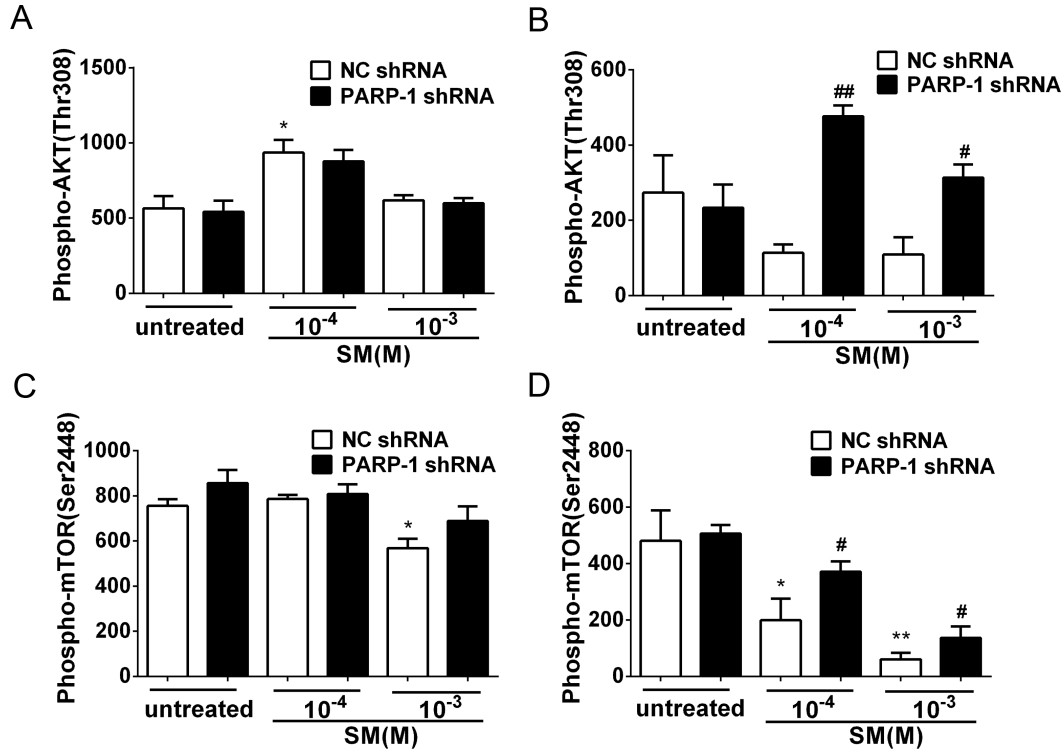

**Figure 8** **SM-induced suppression of Akt/mTOR signaling could be reversed by silencing PARP-1.**
PARP-1-knockdown and control HaCaT cells were treated with 100 μM or 1,000 μM SM. At 6 h and
24 h after exposure to SM, the cells were harvested for the detection of Phospho-AKT (Thr308) (A, B) and
Phospho-mTOR (Ser2448) (C, D). The results are presented as means ± SEM, as determined from three
independent experiments. *$P < 0.05$, **$P < 0.01$, vs. untreated group. #$P < 0.05$, ##$P < 0.01$, vs. SM-
treated group.

## PARP-1 knockdown reversed the SM-induced suppression of the Akt/mTOR pathway

Previously, it was reported that PARP-1 is involved in autophagy induced by DNA damage
through the activation of the key autophagy regulator mTOR (*Munoz-Gamez et al., 2009*).
To investigate whether SM exposure activated mTOR and how PARP-1 participated in
this process, the phosphorylation of Akt and mTOR was evaluated in PARP-1-knockdown
and NC HaCaT cells. The results showed that in the HaCaT cells treated with NC shRNA,
the phosphorylation of Akt and mTOR was suppressed in response to SM stimulation
(Figs. 8B and 8D). At 24 h after exposure of HaCaT cells to SM, PARP-1 knockdown
significantly increased the phosphorylation of Akt and mTOR (Figs. 8B and 8D), whereas
PARP-1 knockdown had no effect on the phosphorylation of Akt and mTOR at 6 h after
exposure to SM (Figs. 8A and 8C). The results suggested that PARP-1 knockdown reversed
the SM-induced suppression of the Akt/mTOR pathway, which might affect autophagy
following SM-induced injury. Taken together, the results suggest that the silencing of
PARP-1 could protect HaCaT cell from SM-induced injury by regulating the apoptosis and
autophagy pathways.

## PARP inhibitor ABT-888 further enhanced the DNA damage after SM exposure

H2AX is a sensitive marker for DNA double-stranded breaks (DSB) (*Fernandez-Capetillo et al., 2004*). When DSB occurs, H2AX is phosphorylated at Ser139 in the nucleosomes surrounding the break point (*Thiriet & Hayes, 2005*). To determine the effect of SM on DNA damage and the potential effect of ABT-888 on preventing SM injury, the phosphorylation of H2AX (S139) was detected in HaCaT cells. HaCaT cells were exposed to 100 µM or 1,000 µM SM before DMEM/F12 (with 10% fetal calf serum) alone or with 10 µM ABT-888 was added. The cells were then analyzed by flow cytometry. The results showed that exposure to 100 µM and 1,000 µM SM increased the phosphorylation of H2AX in HaCaT cells at both 6 h (Fig. 9A) and 24 h (Fig. 9B) following SM exposure. ABT-888 enhanced the phosphorylation of H2AX caused by SM exposure at 24 h (Fig. 9B). This result indicated that PARP inhibitor actually increased the DNA damage after SM exposure.

## DISCUSSION

Due to the multiple actions of PARP-1 on SM-induced injury, the potential use of PARP inhibitors have drawn much attention in SM therapy (*Debiak, Kehe & Burkle, 2009*; *Kehe et al., 2008*; *Rosenthal et al., 2001*; *Steinritz et al., 2007*). In recent years, new generation of specific and potent PARP inhibitors in cancer chemotherapy (*Coleman et al., 2015*; *Gunderson & Moore, 2015*; *Jones et al., 2015*; *Thomas et al., 2007*) provides more selective choices on treating SM injury. ABT-888 is one of the new PARP inhibitor which is over 100 times more potent than the classical PARP inhibitor 3-AB. In this study, we evaluate the effect of PARP inhibitor ABT-888 in HaCaT cells and MEVM treated with SM, and study the mechanism of PARP-1 in SM injury by knockdown of PARP-1 in HaCaT cells.

According to our findings, the application of ABT-888 could reduce pathological damage after severe SM injury (0.64 mg SM/ear) in MEVM, which is consistent with the results of other groups (*Cowan et al., 2003*; *Yourick, Clark & Mitcheltree, 1991*). Besides the reduction on pathological damage, ABT-888 could also reduce edema after severe SM injury in MEVM according to our results, which suggested that ABT-888 may have more therapeutic value in the treatment of skin injuries. Furthermore, we found that ABT-888 showed no protective effect against mild SM-induced injury (0.16 mg SM/ear).

Consistent with the results of the MEVM experiments, we also found that PARP-1 inhibition had a cytoprotective effect on HaCaT cells. ABT-888 significantly increased cell viability at 6 h after severe SM injury (1,000 µM SM exposure), whereas ABT-888 did not protect cell viability under mild SM injury (100 µM SM exposure), suggesting that ABT-888 shows protective effects only at early times after severe SM injury. However, the protective effects could not be observed after 24 h exposure, which is consistent with the results of other groups using the classic PARP inhibitor 3-AB (*Kehe et al., 2008*). The results from our study in MEVM and HaCaT model indicated that the new generation of specific and potent PARP inhibitor ABT-888 had more protective effect than other PARP inhibitors which were used in treatment of SM injury (*Cowan et al., 2003*; *Kehe et al., 2008*).

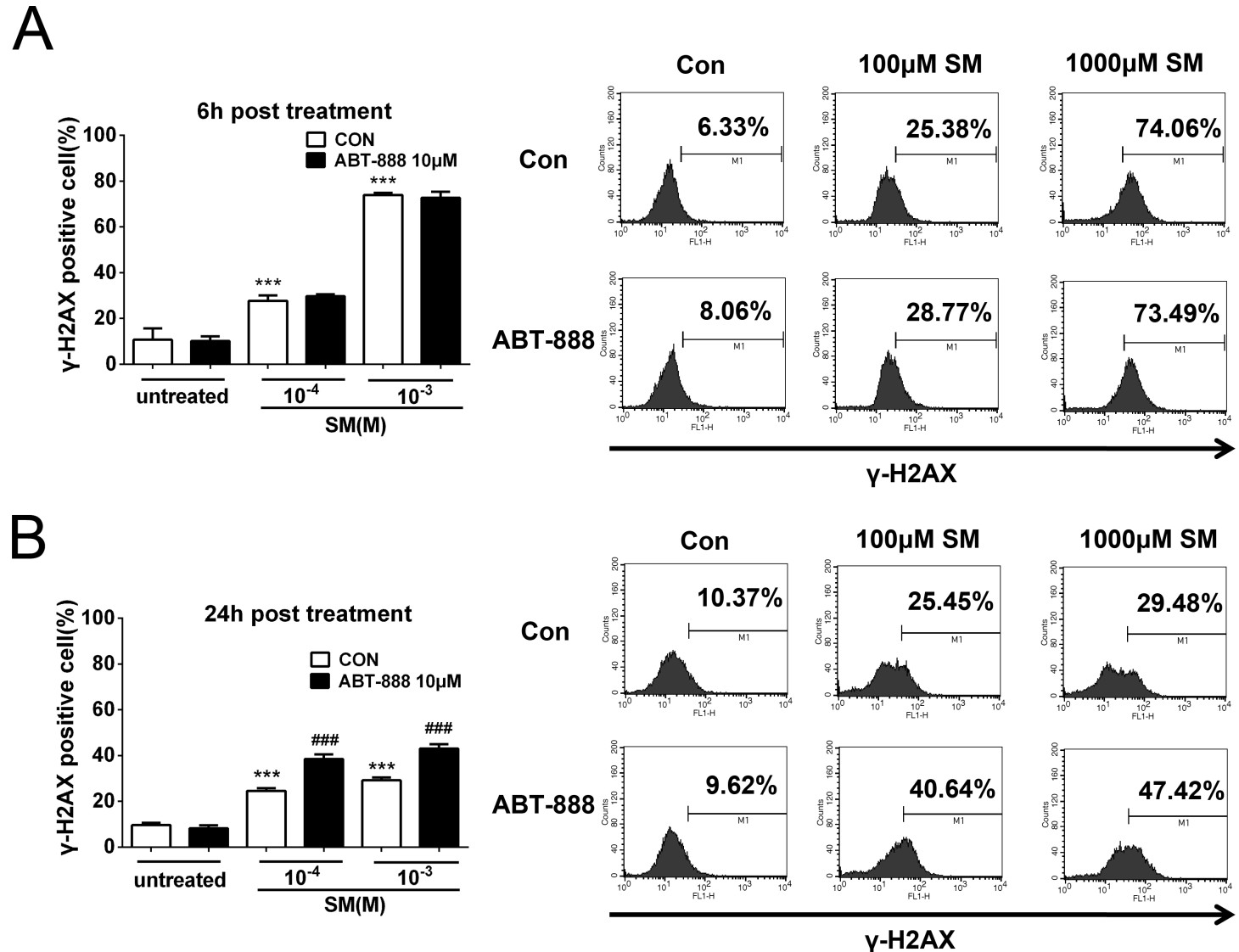

**Figure 9** **SM treatment caused a significant increase in the expression of γ-H2AX, which was further increased by ABT-888.** (A) 6 h post SM exposure, (B) 24 h post SM exposure. Representative examples of FACS data are displayed. ***$P < 0.001$ vs. untreated group. ###$P < 0.001$ vs. SM-treated group. The values are presented as means ±SEM, $n = 6$.

To investigate the reason why the PARP inhibitor showed protective effects after SM injury, we studied the role of PARP-1 in the SM-induced injury in HaCaT cells. Our results showed that the expression of pADPr increased in HaCaT cells after 6 h exposure to either 100 μM or 1,000 μM SM, and the expression of pADPr increased much more in the cells exposed to 1,000 μM SM than those exposed to 100 μM SM. Our results were consistent with other researchers' findings (*Bhat, Benton & Ray, 2006*; *Mangerich et al., 2015*; *Mol, Van de Ruit & Kluivers, 1989*). In addition, our present study demonstrated that SM exposure significantly decreased NAD$^+$ and ATP levels, consistent with the Papirmeister and colleagues' finding that severe SM-induced DNA damage over-activated PARP-1, leading to the depletion of NAD$^+$, the protease release and the blister formation (*Gross et al.,*

*1985*; *Papirmeister et al., 1985*). PARP inhibitor ABT-888 inhibited SM-induced PARP-1 activation. Severe SM injury could cause $NAD^+$ exhaustion, whereas mild SM injury only consumed $NAD^+$. The immediate application of ABT-888 after SM exposure could prevent the $NAD^+$ and ATP reduction after SM exposure.

In addition to its major role in responding to DNA damage, PARP-1 is also an important mediator of apoptotic and/or necrotic pathways (*Aredia & Scovassi, 2014*; *Dantzer et al., 2000*; *Marchenko, Zaika & Moll, 2000*). The cell death pathway involving PARP-1 plays a pivotal role in tissue injury and organ dysfunction in SM-induced toxicity (*Kehe et al., 2008*; *Korkmaz, Tan & Reiter, 2008*). PARP-1 triggers necrotic cell death due to the rapid consumption of the substrate $NAD^+$, leading to ATP depletion (*Los et al., 2002*). In addition, PARP-1 is involved in a caspase-independent apoptosis pathway mediated by apoptosis-inducing factor. Our flow cytometry results showed that SM could induce apoptosis and necrosis in HaCaT cells. ABT-888 significantly decreased the percentage of apoptotic and necrotic HaCaT cells exposed SM. N-methyl-N-nitro-N-nitrosoguanidine (MNNG) is a DNA-alkylating agent. Seong-Woon Yu et al. showed that treatment with MNNG could cause annexin-V staining. Annexin-V staining was not observed in PARP-1–KO fibroblasts (*Yu et al., 2002*), which is consistent with our results that PARP inhibition could reduce the apoptosis induced by alkylating agents.

Caspase-3-catalysed PARP-1 degradation is one of the major events during SM-induced apoptosis. The ability of PARP-1 to repair DNA damage is prevented by the cleavage of PARP-1 by caspase 3/7, a process that plays a central role in the apoptotic pathway and is reported to be involved in SM-induced toxicity (*Debiak, Kehe & Burkle, 2009*; *Mol, Van den Berg & Benschop, 2009*). The specific cleavage of PARP-1 by caspase-3/7 and the generation of 89 and 24 kDa fragments have been extensively used as a biochemical marker of caspase-dependent apoptosis (*Decker & Muller, 2002*). At 6 h after exposure to 1,000 µM SM, western blot analysis revealed marked PARP-1 cleavage into the 89-kDa fragment (c-PARP), while exposure to 100 µM SM showed no PARP-1 cleavage. ABT-888 could decrease c-PARP expression and caspase 3/7 activation at 6 h after exposure to 1,000 µM SM, but showed no effect in cells exposed to 100 µM SM. However, there were no significant protective effects on expression of c-PARP at 24 h after SM exposure.

To further investigate the mechanism of PARP-1 in SM injury, stable PARP-1 knockdown HaCaT cells were constructed using RNA interference. We found evidence that PARP-1 knockdown in HaCaT cell lines resulted in an increase in cell viability after SM exposure, which was consistent with our previous work.

It has been shown that the protective effect of a PARP-1 inhibitor in SM-induced injury is mainly due to a mitigation of the $NAD^+$ depletion caused by SM poisoning (*Hinshaw et al., 1999*; *Meier, Gross & Papirmeister, 1987*; *Mol, Van de Ruit & Kluivers, 1989*; *Smith et al., 1990*). However, PARP-1 is involved in many other important pathological functions, including apoptosis and autophagy (*Virag et al., 2013*), in addition to its important role in cellular energy. PARP-1 is involved in a caspase-independent apoptosis pathway based on the nuclear-to-mitochondrial translocation of pADPr, which triggers the reverse (mitochondrial-to nuclear) translocation of an apoptosis-inducing factor (AIF) (*Wang, Dawson & Dawson, 2009*). The MAP kinase pathway or the PI3-kinase-Akt pathway

have also been shown to be involved in PARP-1 activation (*Aredia & Scovassi, 2014*). Moreover, it has become increasingly clear that p53 is directly regulated by poly(ADP-ribose) polymerase-1 (PARP-1) (*Elkholi & Chipuk, 2014*). To investigate whether the activation of PARP-1 is involved in SM-induced apoptosis, we exposed control and PARP-1-knockdown HaCaT cells to SM and measured the phospho-JNK (Thr183/Tyr185), phospho-p53 (ser46), active caspase 9 (Asp315), active caspase 8 (Asp384), c-PARP (p89) and caspase 3/7 activity. Our data showed that SM activated the apoptosis checkpoints mentioned above and that PARP-1 knockdown suppressed the SM-induced apoptosis checkpoint signals in HaCaT cells, suggesting that PARP-1 knockdown may protect HaCaT cells from SM-induced injury via downregulation of the apoptosis pathway.

Autophagy is a homeostatic "self-eating" pathway that has been conserved among eukaryotic cells (*Klionsky & Emr, 2000*). Autophagy is a defense mechanism for the disposal of damaged organelles or misfolded proteins that protects damaged cells. It is also a type of cell death pathway that induces active cell death (*Navarro-Yepes et al., 2014*). mTOR is a serine/threonine kinase that is highly conserved in all eukaryotes and plays a key role in regulating autophagy. This kinase is a central regulator of cell growth and a major nutrient sensor, and the inhibition of mTOR can induce autophagy (*Kim & Guan, 2015*). The PI3K/AKT pathway is the upstream regulator of mTOR, and the activation of AKT can inhibit autophagy via the regulation of mTOR (*Meijer & Codogno, 2006*). It has been reported that PARP-1 deficiency can reduce autophagy due, in part, to the lack of inhibition of mTOR (*Virag et al., 2013*). However, the effect of PARP-1 deficiency on the AKT/mTOR pathway in SM-induced injury is still unknown. Our results showed that SM exposure could significantly inhibit the activation of the AKT/mTOR pathway and that PARP-1 knockdown, to a certain extent, reversed the toxicity of SM. The results indicated that the activation of PARP-1 may be involved in the inhibition of the AKT/mTOR pathway induced by SM and that inhibition of PARP-1 may inhibit autophagy by re-activating the AKT/mTOR pathway.

In this study, some biological parameters of control and samples treated with the 100 μM SM at 6 h are not significantly different, such as ATP, apoptosis/necrosis cells (%), p-p53, active Caspase 9, active Caspase 8, c-PARP and p-mTOR. This may be the reason why no effect of PARP-1 inhibition is observed in these biological parameters since the effect of SM in cells exposed to 100 μM is not very high.

It should be noticed that PARP inhibitors could enhance DNA damage when applied to cancer cells exposed to DNA damaging agents. A similar phenomenon for the application of PARP inhibitors in SM injury has been reported by *Bhat, Benton & Ray (2006)*; *Bhat et al. (2000)*. In their study, primary keratinocytes displayed rapid phosphorylation and activation of DNA ligase I, a DNA repair enzyme, after SM exposure, and the half-life of activated DNA ligase I increased from 1.3 to 4.3 h in the presence of the PARP inhibitor 3-AB, which implied that the PARP inhibitor could delay the DNA repair in SM injury. Nevertheless, the authors did not provide direct evidence that PARP inhibition could increase DNA damage in SM injury (*Debiak, Kehe & Burkle, 2009*). In our study, we obtained evidence that PARP-1 involved in the repairment of DNA damage induced by SM. H2AX is rapidly phosphorylated following double-strand DNA breaks, which is a

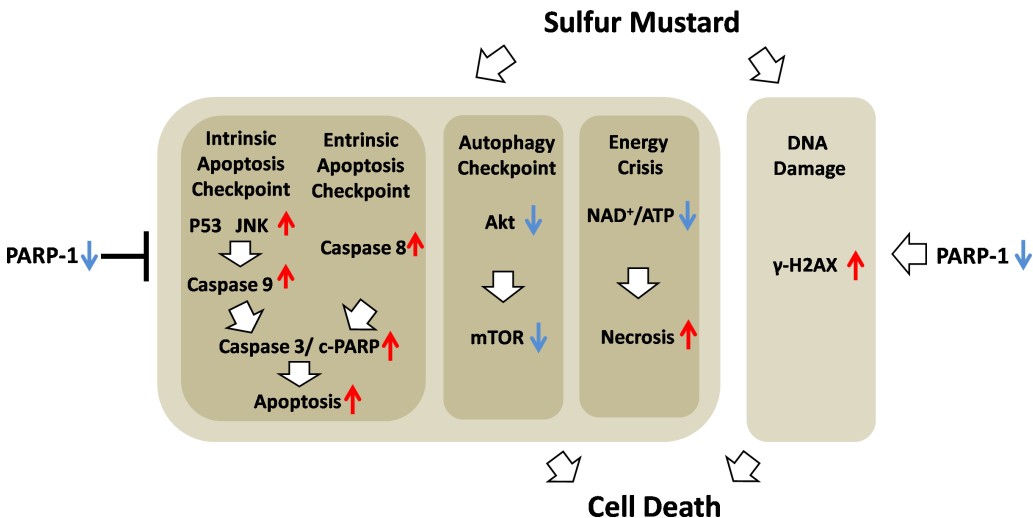

**Figure 10  Schematic of the role of PARP-1 in sulfur mustard injury.**

key step in activating the DNA damage response pathways that are important in repair (*Mah, El-Osta & Karagiannis, 2010*). Our results showed that exposure to either 100 μM or 1,000 μM SM increased the phosphorylation of H2AX (gamma-H2AX) in HaCaT cells. However, SM does not directly induce double-strand breaks (*Kehe et al., 2009*). SM exposure will lead to the formation of unstable adducts that are partly converted into abasic sites and single strand-breaks (*Batal et al., 2014*). Abasic sites and single strand-breaks can then give rise to double strand breaks through indirect mechanisms such as the collapse of replication forks (*Andreassen, Ho & D'Andrea, 2006*). According to our results, ABT-888 further enhanced double strand breaks, which could be explained by the fact that cellular toxicity was decreased by ABT-888 and that more cell divide in the treated samples. Our results indicated that although PARP inhibitor could prevent cell death induced by SM, the increased genotoxicity and likely mutagenesis caused by PARP inhibition should also be considered.

In our study, HaCaT cells were utilized as an *in vitro* cell model to study the toxicity of SM in skin injuries, and discuss the mechanism how PARP-1 inhibitor exerts its efficacy in SM injury. The HaCaT cell line is a spontaneously transformed human epithelial cell line, which is commonly used for skin toxicity testing and the study of SM-induced apoptosis and skin injuries (*Heinrich et al., 2009*; *Kehe et al., 2008*; *Wolf et al., 2015*). But as we know, HaCaT cells have a mutated p53 background (*Lehman et al., 1993*). This may represent a limitation for extrapolation of the present results to human skin cells. However, in the present study we did observe a protective effect of PARP-1 inhibitor in MEVM, indicating that the protective effect of PARP-1 inhibitor could be observed not only in HaCaT cell line but also in normal mouse skin. But still, in order to better reveal the role of PARP-1 in SM induced skin injuries, the effect of PARP-1 inhibitor in SM-exposed primary cultured of human keratinocyte will be investigated in our laboratory in the future study.

## CONCLUSIONS

In conclusion, we demonstrated that ABT-888, a new generation of specific and potent PARP inhibitor, had a therapeutic effect in mouse ear vesicant model (MEVM) and HaCaT cell model after severe SM injury. ABT-888 could reduce SM-induced $NAD^+$/ATP depletion and apoptosis/necrosis in HaCaT cell model. Furthermore, PARP-1 knockdown protects HaCaT cells from sulfur mustard-induced injury by regulating the apoptosis and autophagy checkpoints. Taken together, our results indicated that protective effects of downregulation of PARP-1 in SM injury may be due to the mitigation of apoptosis, necrosis, energy crisis and autophagy. However, PARP inhibitor ABT-888 further enhanced the DNA damage after SM exposure, which indicated that we should be very careful in the application of PARP inhibitors in SM injury treatment (Fig. 10). In addition, the mechanism by which PARP-1 inhibition is protective against SM-induced injury is mainly focused on the mitigation of the $NAD^+$ depletion. However, based on the evidence provided in this article, we have reason to believe that the effect of PARP-1 in the apoptosis and autophagy pathways in sulfur mustard-induced injury cannot be ignored. This study provided a new theoretical basis for PARP-1 inhibition in the treatment of sulfur mustard-induced injuries.

## ACKNOWLEDGEMENTS

We would like to thank Mrs. Xu Xin for the technical assistance.

### Funding

This work was supported by grants from the Chinese Scientific and Technological Major Special Project (2009ZXJ09002-012, 2013ZX09J13103-01B, and 2014ZX09J14103-03A) and the State Key Laboratory of Toxicology and Medical Countermeasures. The funders had no role in study design, data collection and analysis, decision to publish, or preparation of the manuscript.

### Grant Disclosures

The following grant information was disclosed by the authors:
Chinese Scientific and Technological Major Special Project: 2009ZXJ09002-012, 2013ZX09J13103-01B, 2014ZX09J14103-03A.
State Key Laboratory of Toxicology and Medical Countermeasures.

### Competing Interests

The authors declare there are no competing interests.

### Author Contributions

- Feng Liu and Ning Jiang conceived and designed the experiments, performed the experiments, analyzed the data, wrote the paper, prepared figures and/or tables, reviewed drafts of the paper.

- Zhi-yong Xiao, Jun-ping Cheng, Yi-zhou Mei, Pan Zheng, Li Wang, Xiao-rui Zhang and Xin-bo Zhou performed the experiments.
- Wen-xia Zhou conceived and designed the experiments, prepared figures and/or tables, reviewed drafts of the paper.
- Yong-xiang Zhang conceived and designed the experiments.

### Animal Ethics

The following information was supplied relating to ethical approvals (i.e., approving body and any reference numbers):

All experiments were conducted according to the Care and Use Guide for Laboratory Animals by the NIH and were approved by Bioethics Committee of the Beijing Institute of Pharmacology and Toxicology (No. 80-23).

### Data Availability

The raw data is provided as Supplemental Information.

### Supplemental Information

Supplemental information for this article can be found online at http://dx.doi.org/10.7717/peerj.1890#supplemental-information.

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
