# Peer review of "Effects of poly (ADP-ribose) polymerase-1 (PARP-1) inhibition on sulfur mustard-induced cutaneous injuries in vitro and in vivo"

_PeerJ, doi:10.7717/peerj.1890_

## Round 0.1 · original submission · Minor Revisions

Dear authors

Please follow the suggestions of the two expert reviewers when you revise your manuscript.

Reviewer 1 ·

Basic reporting

No Comments

Experimental design

No Comments

Validity of the findings

No Comments

Additional comments

Manuscript No: 8216
Title: Effect of Poly (ADP-ribose) polymerase-1 (PARP-1) inhibition on sulfur mustard-induced cutaneous injuries invitro and invivo.
In this manuscript, the authors described the protective role of PARP inhibitors (ABT-888) to rescue SM-induced skin injury in mouse ear vesicant model (MEVM) and HaCaT cell model. SM is a bifunctional alkylating agent that targets macromolecules like DNA, RNA, proteins, carbohydrates and lipids which finally cause DNA damage, inflammation, apoptotic cell death and necrosis in skin keratinocytes. PARP-1 plays an important role in SM-induced DNA damage repair and can be an important target for therapeutics purposes. Therefore authors described the role of PARP-1 in SM induced skin injury via knockdown of PARP-1 in HaCaT cells which results in downregulation of p-JNK, p-P53, Caspase 9, Caspase 8, c-PARP and Caspase 3 and increased cell viability via regulation of apoptosis, necrosis, energy crisis and autophagy. There are lots of studies available showing the role of PARP in SM-induced skin injury and use of PARP inhibitors as a potential antidote against SM-induced skin injury.
Overall, this is an interesting study where the experiments appear to have been well described, properly conducted, and appropriately interpreted. The presentation of manuscript is clear. However, no mention is made in the text of the results observed with the ABT-888 alone group. The assumption is that there is no effect but this should be added to the manuscript. I would like to suggest rewriting abstract because it is explaining only invitro findings, not the complete experiments like conclusion from MEVM model.

·

Basic reporting

The results are interesting both for the understanding of the effects of sulfur mustard and for the design of novel therapeutic approaches.

Experimental design

The experiments are well designed. The only limitation is the use of the p53 mutated cell line for the study of apoptosis.

Validity of the findings

The results are of quality and relevant to the topic.

Additional comments

The manuscript is an interesting contribution to the understanding of the cellular damage and response induced by exposure to sulphur mustard (SM) and the search for efficient therapies. In particular, the work provides strong evidence for a major role of PARP-1 in SM-mediated cell death thereby confirming an often discussed hypothesis. The interest of the present study lies in the quality of the experiments and the care the authors took at studying each biological endpoints by a combination of biomarkers. In addition, their conclusions on the role of PARP-1 and the therapeutic interest of PARP-1 inhibitors are based on both in vitro and in vivo results. The results thus deserve publication. A few comments have yet to be made:
- A large part of the study deals with the induction of apoptosis by SM and its prevention by treatment with the PARP-1 inhibitor ABT-888. This work was carried out with HaCat keratinocytes which are known to have a mutated p53 background. This represents a strong limitation for extrapolation of the present results to human skin cells. This point should be at least discussed in the text. Even better, a few confirmation experiments on selected endpoints could be made on primary cultures of human keratinocytes.
- In several parts of the manuscript, the authors discuss a “dose dependent response” in HaCat keratinocytes. Such a statement is often an over-interpretation since only two concentrations were applied and that the effects are rarely proportional. Regarding dose-effects, quantification of several endpoints shows a protection by ABT-888 only at the largest concentration. A first explanation could be that involvement of PAPR-1 only occurs above a threshold. Does this seem possible? The author thing the authors may want to check is whether differences between control and samples treated with the lowest concentration are statistically significant. For instance, the toxicity of SM in cells exposed to 100 µM at 6 h is not very high and this may be why no ABT-888 effect is observed.
- In the last part of the work, the authors observed an increase in phosphorylation of H2AX histone and interpret this result as an increase in DNA damage. It should though be reminded that SM does not directly induce double-strand breaks. It rather leads to the formation of unstable adducts that are partly converted into abasic sites and single strand-breaks. The latter lesions could also be produced by the oxidative stress induced by SM. Adducts and SSB can then give rise to double strand breaks through indirect mechanisms such as the collapse of replication forks. It is thus very likely that the increase in DSB observed by the author is explained by the fact that cellular toxicity is decreased by PARP-1 inhibitors and that more cell divide in the treated samples. This does not change the important message put forward by the authors that preventing cell death is at the expense of increased genotoxicity and likely mutagenesis and risk of carcinogenesis.

A few minor corrections should also be made:
- In the abstract, the sentence on Akt/mTOR is not clear.
- Line 46: the work by Yue et al. (Chem Res Toxicol (2014) 27:490-5002014) could be cited
- Line 49 define NAD+ on this line not on line 53
- Line 64 add “s” to “inhibitor” and “effect”
- Line 68 “prevented” rather than “preserved”
- Line 71 “previous” rather than “early”
- Line 76 “available” before “in cancer”
- Line 89 add a reference to the synthesis of ABT-888 and specify the purity
- Line 110-111 what is the surface of the treated skin on the ears?
- Line 135 write “pADPr”. What is the meaning of [10H]?
- Line 167 and others define “BD”
- Line 187 define “PI”
- Line 223 what is the concentration of the SM solution?
- Line 226 ABT-888 is applied before SM-treatment. What happens if it is applied after?
- Line 242-244 better define the % in change. Changes in what?
- Line 430-433 rewrite the sentence which is long and not clear

---

## Round 0.2 · accepted · Accept

The manuscript is accepted for publication.

Reviewer 1 ·

Basic reporting

No Comments

Experimental design

No Comments

Validity of the findings

No Comments

·

Basic reporting

The data are interesting, based on solid experimental evidence.

Experimental design

Experiments are well designed a.nd the results properly analyzed

Validity of the findings

The findings are interesting. They confirm with actual experimental facts hypotheses proposed in the field for many ytears.

Additional comments

The revised version of the manuscript addressed most of the points I raised in the first version. The manuscript can be accepted.